# Crash Injury Severity Prediction Using an Ordinal Classification Machine Learning Approach

**DOI:** 10.3390/ijerph182111564

**Published:** 2021-11-03

**Authors:** Shengxue Zhu, Ke Wang, Chongyi Li

**Affiliations:** 1Jiangsu Key Laboratory of Traffic and Transportation Security, Huaiyin Institute of Technology, Huaian 223003, China; zsx10316@hyit.edu.cn; 2Key Laboratory of Road and Traffic Engineering of the State Ministry of Education, College of Transportation Engineering, Tongji University, Shanghai 201804, China; 1731304@tongji.edu.cn

**Keywords:** crash severity, ordinal classification, imbalance data, machine learning, sampling

## Abstract

In many related works, nominal classification algorithms ignore the order between injury severity levels and make sub-optimal predictions. Existing ordinal classification methods suffer rank inconsistency and rank non-monotonicity. The aim of this paper is to propose an ordinal classification approach to predict traffic crash injury severity and to test its performance over existing machine learning classification methods. First, we compare the performance of the neural network, XGBoost, and SVM classifiers in injury severity prediction. Second, we utilize a severity category-combination method with oversampling to relieve the class-imbalance problem prevalent in crash data. Third, we take advantage of probability calibration and the optimal probability threshold moving to improve the prediction ability of ordinal classification. The proposed approach can satisfy the rank consistency and rank monotonicity requirement and is proved to be superior to other ordinal classification methods and nominal classification machine learning by statistical significance test. Important factors relating to injury severity are selected based on their permutation feature importance scores. We find that converting severity levels into three classes, minor injury, moderate injury, and serious injury, can substantially improve the prediction precision.

## 1. Introduction

The prediction and cause analysis of traffic crashes has always been an important topic for scholars in traffic safety. In the research of this subject, scholars often use statistical methods or machine learning methods to conduct research.

A statistical model usually specifies the mathematical relationship between explanatory variables and crash severity. Based on strict assumptions of uncertainty distribution and hypothesis tests, the statistical model can isolate the effects of explanatory variables on crash severity [1,2]. For example, Cerwick et al. [3] used the mixed logit model and the latent class multinomial logit model to predict crash severity. A large number of crash specific, temporal, roadway, vehicle, driver characteristics, and environmental factors were found significant. Haghighi et al. [4] used standard ordered logit (SOL) and Multilevel ordered logit (MOL) to analyze the effect of roadway geometric features on crash severity. However, statistical models are usually weaker in making predictions than machine learning methods. Iranitalab and Khattak [5] compared the performance of a statistical model, Multinomial Logit (MNL), with three machine learning methods including Nearest Neighbor Classification (NNC), Support Vector Machines (SVM), and Random Forests (RF) in predicting traffic crash severity, and found MNL has the worst prediction accuracy.

Machine learning models are designed to make the most accurate predictions possible. Chang et al. [6] used the Classification And Regression Tree (CART) to predict crash severity, where prediction accuracy is 90.8% for learning data and 91.7% for testing data. Abdel-Aty et al. [7] used a single-layer hidden layer Multi-layer Perceptron (MLP) to predict traffic crash severity with an average prediction accuracy of 73.5%. Delen et al. [8] used an artificial neural network (ANN) to predict the severity of a crash and improved the prediction accuracy. Alkheder et al. [9] used ANN, combined with k-means for clustering, to predict traffic crashes and then compared with probit algorithm to prove that ANN is better than probit in predicting the severity of crashes. Among many methods of crash severity prediction research, neural network methods have better performance and are more popular.

Most existing machine learning methods applied in crash severity prediction treat crash severity levels as nominal data without order information. This unrealistic simplification casts a shadow on machine learning methods’ prediction ability. In general, the severity of traffic crashes is classified as fatal injury or killed, incapacitating injury, non-incapacitating, possible injury, property damage, or no injury. Moreover, the severity of the injury is ordered and increases from no injury to possible injury, to non-incapacitating, to incapacitating injury, and to fatal injury or killed. Between closely related adjacent categories (such as no injury and possible injury), there may be shared unobserved effects or correlations between their data [10]. To the authors’ knowledge, in the field of crash severity study, less attention has been paid to ordinal classification machine learning, and information on natural ordering in injury severity is missed in conventional machine learning, including SVM, decision tree, and MLP. Some statistical models, such as ordered logit and ordered probit [4,11,12], can handle the ordinal severity labels. However, discrete choice models rely on statistical assumptions and pre-defined relationships between severity labels and input variables, which makes them good choices for factor analysis but restricts their prediction accuracy [13].

Gutierrez et al. [14] summarized ordinal classification machine learning algorithms developed to classify categorical variables that show a natural order between the labels. They confirmed that there is no clear winner that performs the best in all possible datasets and problem requirements. The main three categories of ordinal classification machine learning are:(1)Cost-sensitive classification: apply cost-sensitive loss function in the evaluation of the learned system with different costs for different types of misclassification errors. For example, Riccardi et al. [15] proposed cost-sensitive AdaBoost for ordinal regression. The problem of cost-sensitive classification is how to determine the cost matrix without priori knowledge of the ordinal classification.(2)Ordinal binary decomposition: decompose the ordinal target variable into several binary variables, which are then estimated by single or multiple models. Our new ordinal classification method falls in this category. The problem of existing ordinal binary decomposition methods is the violation of rank monotonicity or rank consistency. Related methods and their drawbacks are introduced in the method section later in more detail.(3)Threshold model: extension of the regression model in which distances among the ordered classes are not pre-defined but estimated by finding the optimal thresholds dividing classes [16]. Li and Lin [17] proposed a general reduction framework to transform ordinal regression as a series of binary classification sub-problems and demonstrated that many threshold models and ordinal binary decomposition methods are equivalent.

The number of crash cases in each category is often imbalanced. Usually, the sample size of fatal cases is several times smaller than that of cases in other categories. With imbalanced data, traditional classification algorithms incline to the category with a large amount of data, while the category with a small amount of data is neglected [9]. Many studies merged several minority categories of injury severity into one class and converted multi-class classification problems into two-class (no injury vs. injury) classification problems [8]. Another option is to turn the multi-class classification problem into a three-class (no injury, minor injury, and fatal injury) problem [3,6]. Some studies have tried to deal with imbalanced data by under-sampling majority class examples [18] and by oversampling minority class examples [19,20] and achieved good results. 

Although some scholars have tried to combine several injury severity levels into fewer categories [8], there will still be a class-imbalance problem, and the predicted results will still incline towards the category of large proportion. This paper focuses on different combinations of severity categories that can relieve imbalance and keep the model’s ability to predict crash injury severity. SMOTE-NC (Synthetic Minority Oversampling Technique for Nominal and Continuous) is applied to oversample the minority class. We compare the performance of three classifiers: MLP, XGBoost, and SVM. The best classifier is combined with an ordinal binary decomposition method to handle ordinal crash severity labels.

The aim of this paper is to propose an ordinal machine learning classification approach that overcomes the ordinal nature of crash severity data and class-imbalance problems. The contributions of this presented approach include:(1)To the authors’ knowledge, this is the first paper applying ordinal classification machine learning to predict traffic crash injury severity using real-world crash data.(2)We propose an ordinal classification machine learning method that satisfies rank monotonicity and rank consistency and takes advantage of probability calibration and the movement of optimal probability threshold to generate superior classification results compared to existing ordinal classification algorithms.(3)We test six severity category-combination strategies and find the best three-class combination plan.

The rest of this paper is constructed as follows. The second section describes and analyzes the characteristics of crash data. The third section presents the research methods involved in this paper, including sampling, severity category-combination, machine learning, and ordinal classification. The fourth section shows the comparison and analysis of the results. The conclusions of this paper are included in the fifth section.

## 2. Data Description

The data were collected from the Highway Safety Information System (HSIS) for crashes that occurred in California in 2010. Variables in the crash dataset include those related to intersections, road segments, and historical traffic crashes. Several variables were dropped when the null value occurred too frequently in the dataset. The California traffic crash data contains three data files, the crash file, vehicle file, and occupant file. The crash file contains 52 variables such as time, location, crash severity, the total number of injuries, weather, etc. The vehicle file contains 42 variables such as vehicle model, whether the driver makes a phone call, whether the driver is drunk, etc. The occupant file contains ten variables such as age, gender, the severity of injury, type of collision, etc. These three data files were merged according to the crash number and the crash vehicle number. The data contains 104 variables observed in one crash, including the injury severity of the person involved. In this dataset, five crash injury severity levels are defined, namely non-injury crash (denoted as NIC), complaint of pain (denoted as COP), other visible injury (denoted as OVI), severe injury (denoted as SI), and killed (denoted as KSI). The severity level of the crash-related personnel injury is shown in Figure 1. The frequencies of severity levels in the data were 80,474 (57.20%), 41,642 (29.60%), 15,200 (10.80%), 2714 (1.93%), and 660 (0.47%), respectively.

We select 17 major influencing variables in this study: occupant type, seating position, type of collision, primary collision factor, first associated factor, roadway class, ejected, object struck, the total number of vehicles involved, alcohol involved, driver’s gender, driver’s age, occupant’s age, vehicle year, motorcycle involved, driver’s safety equipment, and occupant’s safety equipment. Primary collision factor is the one element that best describes the cause of the collision or, if removed, would have prevented the collision from occurring. First associated collision factor is the most important one of factors or violations that contributed, but were not the main cause of the collision. There are 14 categorical variables, except for driver’s age, occupant’s age, and vehicle model year. The Appendix A provides the descriptive statistics of these variables. In total, there are 140,690 crash records, out of which 139,555 remained after cleaning missing data. Samples with missing values were simply removed because the proportion of samples with missing values is relatively small. Other errors were not found in the Highway Safety Information System (HSIS) dataset.

## 3. Methodology

We summarize the research framework as a flowchart in Figure 2.

After data cleaning, we try six methods of category combination that merge crash injury severity levels into fewer categories. SMOTE-NC oversampling is applied to relieve the class-imbalance problem. We compare SVM, XGBoost, and MLP, and choose the best one as the classifier used in the ordinal classification method. The permutation feature importance is also analyzed with the chosen classifier. 

The proposed ordinal classification method contains five main steps. First, the label to predict (crash injury severity) is decomposed with one-vs-all binary decomposition. Second, a multiple-output classifier or multiple single-output classifiers (chosen from SVM, XGBoost, and MLP) are trained to predict crash injury severity. Third, the predicted probabilities are calibrated to remove the bias from the classifier and data sampling. Fourth, the cumulative probabilities are calculated based on calibrated probabilities, which satisfies both the rank monotonicity and rank consistency. Fifth, the threshold moving method can help to find the optimal threshold that converts probabilities into crash injury severity predictions.

All classifiers trained in this paper are established using Python programming language with supported libraries, including keras, tensorflow, xgboost, hyperopt, and scikit-learn. The computing platform is a desktop computer with an AMD Ryzen 1700X 8-core processor and Windows 10 operating system. The proposed ordinal classification method, along with other methods, is evaluated through cross-validation. The performance of each method is compared and tested by a statistical significance test. The computational cost of the proposed method is discussed to show its computation efficiency.

### 3.1. Imbalanced Data Preprocessing

One of the critical characteristics of the crash dataset is that the number of crashes leading to death and severe injury is always much less than those of trivial injury. The problem of imbalanced data is prevalent in the field of traffic accident studies. For example, in our data NIC accounts for 57.20% of all crashes. This imbalance of data means that a dummy classifier that classifies all instances to NIC would still achieve an accuracy score of 57.20%. This issue would have a detrimental effect on the training process. Classifiers such as artificial neural networks, support vector machines, and decision trees are designed for balanced data with a roughly equal sample size of each class. In the case of imbalanced data, classifiers tend to overly focus on the class with the largest proportion and ignore the minority class. However, accurately predicting the minority class, SI and KSI in this case, is the main purpose of machine learning training. Existing research has tried to combine the categories of severe injuries in traffic crashes and turn multi-class problems into a binary-class problem to make predictions. However, training a binary-class classifier limits the model’s ability to distinguish different levels of injury severity and therefore reduces the model’s practical value.

This research performs oversampling and category combination to solve the problem of imbalanced data. We combine the five crash severity levels into three classes in order to reduce the difficulty in severity prediction. We propose all six possible ways of category combination that can convert five crash severity levels into three classes while keeping an ordinal nature (illustrated in Figure 3). Each class contains at least one of the five crash severity levels, and the severity levels are exclusive over classes. For all combinations, NIC is always included in class 1, and KSI is always included in class 3. All instances in class 3 have higher severity levels than instances in class 2, and all instances in class 2 have higher severity levels than instances in class 1. The difference between combinations is how COP, OVI, and SI are assigned into classes. As shown in Figure 4, the proportion of each class in the traffic crash data is still uneven in each combination, but the imbalance is relieved compared to the original five categories. In addition, the 3-class classification problem has more explanatory ability compared to the 2-class problem. It can be interpreted that the five severity categories are combined into three new classes: minor injury crashes, moderate injury crashes, and serious injury crashes.

Another important task we need to complete is oversampling of the minority class. At present, there are two main sampling methods to deal with the imbalance-data problem, namely under-sampling and oversampling. Machine learning algorithms are data-hungry and require extensive data for model training, making under-sampling unpreferable, especially when the minority class sample size is small. Because the dataset is a mix of categorical and continuous features, this research uses SMOTE-NC sampling, a variation of SMOTE sampling. SMOTE-NC creates synthetic data for categorical as well as continuous features in the data set. SMOTE-NC treats categorical features differently from continuous features. For continuous features, SMOTE-NC sampling is an interpolation algorithm that looks for features between the data sample and supplements data with similar characteristics for minority class instances. By contrast, the categorical variable’s value of a newly generated sample is decided by picking the most frequent category of the nearest neighbors present during the generation [21]. The proportion of each class in each combination after oversampling is shown in Figure 5. Class 1 is the majority class in all six combinations. Classes with an instance number smaller than 20% of class 1 are chosen to be oversampled. For all combinations except combination 3, class 3 is oversampled. For combination 6, class 2 is also oversampled. SMOTE-NC sampling is performed on class 3 in combination 1, 2, and 4–6, and on class 2 in combination 6. The sampling rate ensures that the minority class instance size equals one-fifth of the majority class instance size. The new proportion of classes in each combination after sampling is shown in Figure 5. Since classes 2 and 3 are more than one-fifth of class 1 in combination 3, no sample in combination 3 is oversampled.

### 3.2. Ordinal Classification

#### 3.2.1. Cumulative Binary Decomposition

Given a dataset $D={\left\{x_{i},y_{i}\right\}}_{i=1}^{N}$ with the class label $y_{i}\in \left\{C_{1},C_{2},\cdots ,C_{K}\right\}$ where $C_{1}<C_{2}<\cdots <C_{K}$, as expressed in (1), the cumulative binary decomposition method encodes $y_{i}$ into K-1 binary labels $y_{i}^{\left(1\right)},y_{i}^{\left(2\right)},\cdots ,y_{i}^{\left(K-1\right)}$ that $y_{i}^{\left(k\right)}=1\left\{y_{i}>C_{k}\right\}$. The indicator function $1\left\{y_{i}>C_{k}\right\}$ is 1 when $y_{i}>C_{k}$ and 0 when $y_{i}\leq C_{k}$.
(1)$$y=\left[\begin{array}{c} C_{1}\\ C_{2}\\ \vdots \\ C_{K} \end{array}\right]\Rightarrow \left[\begin{array}{cccc} 0 & 0 & \cdots & 0\\ 1 & 0 & \cdots & 0\\ \vdots & \vdots & \cdots & 0\\ 1 & 1 & \cdots & 1 \end{array}\right]$$

A single multi-output model or *K*-1 single-output models can be trained with the binary decomposed dataset ${\left\{x_{i},y_{i}^{\left(k\right)}\right\}}_{i=1}^{N}$ where k∈{1,2,⋯,K−1}. After training, the predicted probability of $y_{i}^{\left(k\right)}$ is essentially the predicted cumulative probability $\hat{p}\left(y_{i}>C_{k}\right)$. Frank’s method and Cheng’s method are both based on $\hat{p}\left(y_{i}>C_{k}\right)$.

Frank’s method [22] first calculate the probability of each class based on (2), and the predicted class is given by (3).
(2)$$\hat{p}\left(y_{i}=C_{k}\right)=\{\begin{array}{c} 1-\hat{p}\left(y_{i}>C_{k}\right),\;k=1\\ \hat{p}\left(y_{i}>C_{k-1}\right)-\hat{p}\left(y_{i}>C_{k}\right),\;k\in \left\{2,\cdots ,K-1\right\}\\ \hat{p}\left(y_{i}>C_{k-1}\right),\;k=K \end{array}\;$$
(3)$$\underset{k}{{\hat{y}}_{i}=\mathrm{argmax}}\left[\hat{p}\left(y_{i}=C_{k}\right)\right]\;$$

Cheng’s method [23], by contrast, predicts class labels by (4).
(4)$${\hat{y}}_{i}=\sum_{k=1}^{K-1}1\left\{\hat{p}\left(y_{i}>C_{k}\right)>0.5\right\}+1$$

#### 3.2.2. One-vs-All Binary Decomposition

Different from cumulative binary decomposition, one-vs-all binary decomposition method encodes $y_{i}$ into *K* binary labels $y_{i}^{\left(1\right)},y_{i}^{\left(2\right)},\cdots ,y_{i}^{\left(K\right)}$ that $y_{i}^{\left(k\right)}=1\left\{y_{i}=C_{k}\right\}$, which is expressed in (5).
(5)$$y=\left[\begin{array}{c} C_{1}\\ C_{2}\\ \vdots \\ C_{K} \end{array}\right]\Rightarrow \left[\begin{array}{cccc} 1 & 0 & \cdots & 0\\ 0 & 1 & \cdots & 0\\ \vdots & \vdots & \ddots & 0\\ 0 & 0 & \cdots & 1 \end{array}\right]$$

A single multi-output model can be trained to predict $\hat{p}\left(y_{i}=C_{k}\right)$ and guarantee that
(6)$$\sum_{k=1}^{K}\hat{p}\left(y_{i}=C_{k}\right)=1$$

Beckham and Pal [24] propose a method that
(7)$${\hat{y}}_{i}=\sum_{k=1}^{K}\beta_{k}\cdot \hat{p}\left(y_{i}=C_{k}\right)\;$$
where $\beta_{k}$ is to be determined. One option is setting $\beta_{k}=k$; another is to calculate $\beta_{k}$ by optimizing the following objective function (8). We denote these two options as Beckham1 and Beckham2.
(8)$$\underset{\beta_{k}}{\max}\sum_{i=1}^{N}{\left[y_{i}-\sum_{k=1}^{K}\beta_{k}\cdot \hat{p}\left(y_{i}=C_{k}\right)\right]}^{2}\;$$

#### 3.2.3. Existing Drawbacks

Both cumulative binary decomposition and one-vs-all binary decomposition have drawbacks. 

Rank monotonicity requires that $\hat{p}\left(y_{i}>C_{k}\right)$ ≤ $\hat{p}\left(y_{i}>C_{j}\right)$ for any *k* > *j*. However, predicted cumulative probabilities based on cumulative binary decomposition do not guarantee rank monotonicity [25] since cumulative probabilities $\hat{p}\left(y_{i}>C_{k}\right)$ and $\hat{p}\left(y_{i}>C_{j}\right)$ are predicted independently. If $\hat{p}\left(y_{i}>C_{k}\right)>\hat{p}\left(y_{i}>C_{j}\right)$ for any *k* > *j*, then $\hat{p}\left(C_{k}\geq y_{i}>C_{j}\right)<0$, which is unrealistic and hurts model performance.

Predicted class probabilities based on one-vs-all binary decomposition do not guarantee rank consistency, which requires $p_{i}\left(k\right)=\hat{p}\left(y_{i}=C_{k}\right)$ to have a convex shape, illustrated in Figure 6. 

#### 3.2.4. Proposed Method

We summarize all methods in Table 1. Frank’s method and Cheng’s method are both based on cumulative binary decomposition. Frank’s method converts predicted cumulative probabilities into class probabilities, while Cheng’s method applies cumulative probabilities directly to predict class labels without knowing class probabilities. Beckham1 and Beckham2 are based on one-vs-all binary decomposition and class probabilities. 

We propose a new method that uses one-vs-all binary decomposition and *K* single output models to predict class probabilities. The main difference between our method and Beckham1/Beckham2 is that we convert predicted class probabilities into cumulative probabilities:(9)$$\hat{p}\left(y_{i}>C_{k}\right)=\sum_{j=k+1}^{K}\hat{p}\left(y_{i}=C_{j}\right)$$

The advantage of this method is that it generates predicted cumulative probabilities satisfying rank monotonicity. Then the class label is determined as follows:(10)$${\hat{y}}_{i}=\sum_{k=1}^{K-1}1\left\{\hat{p}\left(y_{i}>C_{k}\right)>T_{k}\right\}+1$$
where *T_k_* is the optimal threshold of $y_{}^{\left(k\right)}$. Some machine learning algorithms, such as tree-based learning, usually generate biased probabilities. Probabilities can also be distorted because of data imbalance and sampling [26]. Therefore, we find the F1-maximizing *T_k_* with validation data instead of simply setting *T_k_* to 0.5.

Since $\hat{p}\left(y_{i}=C_{j}\right)$ in (6) could be biased, the bias is delivered to $\hat{p}\left(y_{i}>C_{k}\right)$ and causes inaccurate estimation of ${\hat{y}}_{i}$ in (7). We perform probability calibration by isotonic regression to remove bias in $\hat{p}\left(y_{i}=C_{j}\right)$. There are two main methods to calibrate probability: Platt scaling and isotonic regression. Platt [27] introduced Platt scaling, which trains a logistic regression to map the original output to the real class probability. Isotonic regression is a non-parametric approach introduced by Zadrozny and Elkan [28,29]. Isotonic regression is preferable to Platt scaling when the sample size is large enough. Isotonic regression fits a piecewise constant non-decreasing function, where predicted probabilities or scores in each bin are assigned the same calibrated probability that is monotonically increasing over bins. More formally,
(11)$$\begin{array}{ll} \underset{\theta ,a}{\min} & \sum_{m=1}^{M}\sum_{i=1}^{N}1\left(a_{m}\leq {\hat{p}}_{i}<a_{m+1}\right){\left(\theta_{m}-y_{i}\right)}^{2}\\ \mathrm{subject}\;\mathrm{to}\; & 0=a_{1}\leq a_{2}\leq \cdots \leq a_{M+1}=1,\;\theta_{1}\leq \theta_{2}\leq \ldots \leq \theta_{M} \end{array}$$
where *M* is the number of bins, $a_{1},a_{2},\cdots ,a_{M+1}$ are the interval boundaries, $\theta_{1},\theta_{2},\ldots ,\theta_{M}$ are the corresponding calibrated probabilities for that falls in each bin.

### 3.3. Machine Learning Algorithms

We test the performance of three classifiers, Multi-Layer Perceptron (MLP), eXtreme Gradient Boosting (XGBoost), and Support Vector Machine (SVM), on 5-category classification. The winner of three candidates is kept for a 3-class problem and other analysis tasks after.

MLP is a pass-forward artificial neuron network that maps an n-dimensional input vector to an m-dimensional output vector. It has many successful applications in classification tasks such as MNIST handwriting digit number recognition by transforming high dimensional input of related elements to low dimensional discriminative representation. Several researchers have attempted to utilize deep learning frameworks to model potential factors that may lead to different injury severity levels [30,31]. They input the randomly shuffled dataset directly into the network to capture the feature of all input factors of a particular crash. 

Back-propagation multi-layer perceptron (BP-MLP) applies weighted input from every previous layer to a non-linear function, evaluates the difference between network output and actual label, and optimizes the parameters in the network using optimizers such sophistic gradient decedent (SGD) or Adam optimizer. Thus, the characteristic of a particular MLP model can be defined by its depth, non-linear function of each layer, loss function, and optimizer. This research utilizes two hidden layers in the network. The numbers of neurons are 64 and 10, respectively. Both layers use a rectifier linear unit (ReLU) as the activation function. We then feed the mapped 3-dimension output learned representation vector to a softmax layer to compute the final predicted class and predict probabilities for the input variables.

Among the 17 variables used to predict injury severity, there are 14 categorical variables, except driver’s age, occupant’s age, and vehicle model year. When training a neural network, one-hot encoding is more appropriate for categorical data where no specific numerical relationship exists between categories. This involves representing each categorical variable with a group of binary vectors that has one {0,1} code for each unique variable value. However, One-hot encoding dramatically increases the dimension of data. For example, if a {0,1} code is used to represent every numerical code of object1, then the one-hot encoding will create 99 more dimensions than numerical encoding. One-hot encoding also leads to sparse data space, making it challenging to optimize neural networks.

XGBoost is a Gradient Tree Boosting-based algorithm that has been proven to be a powerful classifier. The advantage of XGBoost in this study is that decision tree-based machine learning has no issues with the numerical encoding of categorical variables. Moreover, XGBoost requires much less training time than neural network and often produce remarkable prediction results in crash-related studies [32,33,34].

SVM has been and still is a widely used classifier. Many studies of traffic crash injury prediction have applied SVM as a benchmark classifier [30,35,36]. Therefore, it is used as such in this study.

To extract the maximum performance out of classifiers, we need hyperparameter tuning to determine the optimal combination of hyperparameters. Hyperopt is one of the most popular hyperparameter tuning packages and implements the Tree of Parzen Estimators (TPE) algorithm to search the optimal value of hyperparameters efficiently in a search space described by the user [37]. We apply Hyperopt in this paper to optimize the main hyperparameters of MLP, XGBoost, and SVM. The hyperparameters of MLP include the number of layers/neurons, activation function in each layer, optimizer, learning rate, number of epochs, and batch size. The hyperparameters of XGBoost include the number of estimators, learning rate, maximum depth, subsample ratio, etc. The hyperparameters of SVM are the C parameter and gamma. 

### 3.4. Cross-Validation and Evaluation Metrics

We use stratified 10-fold cross-validation to evaluate the classification algorithm’s performance. Stratified 10-fold cross-validation divides the 139,555 records randomly into ten equal-sized subsets. Each subset has the same proportion of each class as the total dataset. At each time, eight subsets are used for sampling (if required) and training, and one subset is used for probability calibration and threshold optimization (only for the ordinal classification method proposed in this paper). The last subset is used to test the performance of the trained model. This process rotates through each subset, and the average precision, recall, and F1 score of each class represent the algorithm’s performance.

### 3.5. Statistical Significance Test

Machine learning algorithms are commonly evaluated using k-fold cross-validation, and their evaluation metrics, such as mean accuracy scores, are compared directly. Statistically significance tests are designed to test whether the difference between evaluation metrics is statistically significant or the result of a statistical fluke. The null hypothesis is that metric scores observed from two algorithms were drawn from the same distribution. If this assumption is rejected, it suggests that the difference in metric scores is statistically significant. Otherwise, the two algorithms’ performances are statistically equal.

K-fold cross-validated paired Student’s *t*-test is the most used statistical test for machine learning algorithms comparison. However, the calculation of the t-statistic in the test is misleading since the metric scores in each sample are not independent [38]. In k-fold cross-validation, a given observation will be used in the training dataset k-1 times. This means that the estimated metric scores are dependent.

Dietterich [38] recommended a resampling method called 5 × 2 cross-validation that involves five repeats of 2-fold cross-validation. Two-fold cross-validation can ensure that each observation appears only in the train or test dataset once. A paired Student’s *t*-test is used on the results.
(12)$$t=\frac{\mu }{\sqrt{\frac{1}{5}\sum_{i=1}^{5}\left({\left(\Delta_{i}^{\left(1\right)}-\mu \right)}^{2}+{\left(\Delta_{i}^{\left(2\right)}-\mu \right)}^{2}\right)}}$$
where:$\Delta_{i}^{\left(1\right)}$ is the scores difference of two algorithms for the first fold of the *i*-th 2-fold cross-validation;$\Delta_{i}^{\left(2\right)}$ is the scores difference of two algorithms for the second fold of the *i*-th 2-fold cross-validation;$\mu =\frac{\Delta_{1}^{\left(1\right)}+\Delta_{1}^{\left(2\right)}}{2}$ is the mean of scores difference for the first 2-fold cross-validation.

Under the null hypothesis that two algorithms are statistically equal, *t* is assumed to follow a Student’s t-distribution with 5 degrees of freedom. If *t* stays close enough to 0, then the null hypothesis is satisfied. The threshold is 2.571 at the 95% confidence level. 5 × 2 cross-validation is used in this paper to compare algorithms’ performance.

## 4. Results

### 4.1. Comparison of Classifiers

The result of the 5-category classification problem is listed in Table 2. We compare each classifier’s precision rate and find that XGBoost has the highest precision rate in COP, SI, and KSI categories. MLP only outperforms other classifiers in category OVI. The gap between the performance of XGBoost and MLP may be caused by the data characteristic that most variables are categorical. Therefore, we utilize XGBoost as the only classifier used for analysis in the following sections.

The performance of SVM relies on marginal data that lies near the separating hyperplane. SVM yields poor performance when fed with data with ambiguous distinction or imbalanced class. Several modifications to the SVM kernel function and preprocessing methods have been used to improve SVM’s capability to distinguish minority samples. This paper uses radial basis function (RBF) as SVM’s kernel function and achieves significant improvement on minority class prediction compared to other kernel functions. Although SVM can identify 100% NIC instances, it is still the worst classifier and fails to identify any SI and KSI instances. 

In general, the precision of injury severity decreases as the severity level rises. More than 99% of NIC cases can be correctly classified regardless of the classifier used. For COP, the precision is about 60%, but the precision of SI and KSI decreases dramatically to almost 0. As discussed in the Introduction, the poor performance on serious injury crashes is caused by class-imbalance.

### 4.2. Comparison of Category-Combination and Sampling

In order to overcome the shortcomings of the five-category problem, this research proposes six ways of category-combination and generates a 3-class problem through category-combination. As shown in Table 3, it is clear that the macro-average precision rate is improved for most combinations except combination 6. However, it can be seen that the precision rate of class 3 (KSI only) is still very low in combinations 1 and 4 because KSI is not combined with others to relieve the imbalance.

In Table 4, after preprocessing by SMOTE-NC, the precision rate of each class is further improved. Among all combinations, combination 3 has the highest F1 score of 45.0% for class 3, but it combines OVI with SI and KSI and loses the ability to predict serious injury crashes. Combination 5 has the second-highest F1 score of 24.5% for class 3, but its F1 score for class 2 is only 19.4%. Combination 2 achieves acceptable F1 scores of 90.1%, 78.3%, and 24.1% for classes 1, 2, and 3, respectively. Moreover, combination 2 groups COP and OVI into class 2 and groups SI and KSI into class 3, which is a reasonable combination strategy. The three classes can be considered as minor injury crashes, moderate injury crashes, and serious injury crashes. Therefore, we apply combination 2 to convert 5-category into a 3-class classification problem.

### 4.3. Feature Importance

After category-combination and SMOTE-NC sampling, the classification model is more efficient in predicting the severity of crash injuries than the original five-category problem. We analyze the permutation feature importance of the classification models, as shown in Table 5. In combination 1-3, occupant type has the most significant impact on the injury severity to crash-injured individuals. Ejected from vehicle has the second-highest importance in combinations 3 and 5. In combination 1 and 2, the number of vehicles involved in the crash has the second-greatest impact on the severity of injuries to crash-injured individuals. Vehicle model year is also an important feature in combinations 3, 4, and 5. Based on repeated permutation feature importance calculation, we find that the differences between the most and less important input features are statistically significant.

It is worth noting that in each combination, features with high importance are basically the same. They are ejected from vehicle, number of vehicles, occupant type, and vehicle model year. As shown in Appendix A, ejected from vehicle and occupant type are highly related to the severity of the injury. The driver’s injury severity is more considerable when the number of vehicles involved is one or two. The proportion of cases in which drivers were ejected from vehicles is not particularly large, accounting for only 2.45% of the data. In cases where drivers were ejected from vehicles, the proportion of fatal and severe injuries is high, accounting for 27–29%. In cases where drivers were not ejected from vehicles, the ratio of fatal and severe injury is only 1.7%.

### 4.4. Comparison of Ordinal Classifications

In total, we test the performance of 5 ordinal classification methods on injury severity data, including Frank’s method, Cheng’s method, Beckham1, Beckham2, and the method proposed by this paper. We also compared the results of ordinal classification methods with nominal classification to prove ordinal classification’s advantage. In each method, XGBoost is used as the basic classifier.

In Section 1, we explained why ordinal classification methods are better than nominal classification when the labels are ordinal. In Section 3, we interpreted the drawbacks of ordinal classification benchmarks used in this paper and why our proposed ordinal classification method is more advanced theoretically. We believe that our proposed method should outperform other ordinal classification methods, which outperform nominal classification. Most results shown in Table 6 are consistent with our expectations.

The ordinal classification method proposed in this paper achieves the highest precision rate for class 3, at 41.2%. The corresponding recall rate is acceptable, at 21.3%. The F1 score is 28.1%, which is also the highest among all methods. In the meantime, the proposed approach can still get high F1 scores for classes 1 and 2. This method also has the highest macro-average precision and the third-highest macro-average F1 score. 

As expected, Frank’s method and Cheng’s method have the second highest and third highest F1 score for class 3. Cheng’s method gets almost the same F1 scores for classes 1 and 2 as Frank’s method. This shows that Frank’s and Cheng’s methods are superior to the traditional nominal classification method, although rank monotonicity is not satisfied. 

Surprisingly, Beckham1 and Beckham2 perform worse than nominal classification. Beckham1′s F1 scores for class 2 and 3 are smaller than these of nominal classification. Beckham2 cannot even predict any cases in class 3, resulting in a 0% F1 score for class 3. A possible reason is that Beckham’s method is adversely impacted by the class-imbalance issue. Beckham’s method relies on the estimation of β_k_, which could be biased if the numbers of cases in classes are not equal-sized.

5 × 2 cross-validation and paired Student’s *t*-test are used to test whether the ordinal classification method proposed in this paper is statistically better than other methods. As shown in Table 7, we compare this paper’s method (method A) and other methods (method B) by testing whether their accuracy scores are from the same distribution. All p-values are smaller than 0.01, indicating that performance differences are statistically significant, and this paper’s method is superior to the nominal classification method and other ordinal classification methods.

The computational cost of the proposed method is not much higher than nominal classification and other ordinal classification methods. The main computational cost of classification, either categorical or ordinal, is training k or k-1 single output XGBoost classifiers, which takes 21.8 s on the computing platform. Compared to other methods, the extra computational work of the proposed method is probability calibration and threshold moving, which costs about 4.5 s and are much faster than XGBoost training. Therefore, the proposed method can improve model performance with minimal extra computational cost.

Figure 7 and Figure 8 present the predicted probabilities before and after calibration, respectively. The probability plot is a standard way to check how predicted probabilities fit empirical probabilities. Take class 1, for example, in which all samples are binned into groups based on their predicted probabilities of class 1. For each bin, we calculate the percentage of samples that are actually in class 1 (fraction of positives). The horizontal axis of Figure 7 and Figure 8 are the mean predicted probabilities of each bin, and the vertical axis is the corresponding fraction of positives. Perfectly calibrated probabilities should have the mean predicted probability equal to the fraction of positives in each bin and should form a diagonal line in the probability plot.

Before calibration, the predicted probabilities of class 1 are very close to being perfectly calibrated. The predicted probabilities of class 2 are slightly underestimated. For example, when the predicted probability of class 2 is around 60%, the actual fraction of positive cases is 80%. This underestimation bias could be caused by XGBoost itself since decision tree-based classifiers do not generate calibrated probabilities. The predicted probabilities of class 3 are obviously overestimated since the probability plot is below the perfectly calibrated line. This problem is due to class imbalance and oversampling, which distorts the class distribution in the original data. Therefore, if the biased and uncalibrated probabilities of classes 1, 2, and 3 are directly used to calculate the cumulative probabilities, the cumulative probabilities will also be wrong and unreliable.

After isotonic regression, all predicted probabilities are perfectly calibrated, as shown in Figure 8.

The thresholds *T_k_* in (7) are determined by finding the F1-maximizing thresholds for the validation data. The optimal thresholds found for the data used in this paper are *T*_1_ = 0.43 and *T*_2_ = 0.33. Since 0.5 is the default threshold used in many studies and algorithms, we compare the optimal thresholds to the default threshold in Table 8. The default threshold leads to significantly worse results than the optimal thresholds. For the default threshold, the precision rate of class 3 is 15.8%, much smaller than 41.2% of the optimal thresholds, and the F1 score of class 3 is also smaller than that of the optimal thresholds.

## 5. Conclusions

This research proposed an ordinal classification machine learning method to improve the prediction of imbalanced traffic crash injury severity. SMOTE-NC oversampling and category-combination are applied to relieve the class imbalance problem. XGBoost, SVM, and multi-layer perceptron machine learning are utilized to predict the injury severity of traffic crashes. Based on the analysis results, the effects of ejected from vehicle, number of vehicles involved, occupant type, and vehicle model year on the severity of traffic crashes are found to be important. The experimental results suggest that the proposed ordinal classification method provides better prediction results than other existing ordinal classification methods and traditional nominal classification, especially in minority classes. It was shown from the results that probability calibration and optimal thresholds are helpful in injury severity prediction.

Future efforts should focus on the following aspects: (1) establish a more comprehensive ordinal classification that combines cost-sensitivity with the ordinal classification method proposed in this paper. (2) try to solve the 5-category classification problem without combining any two or more categories. 

## Figures and Tables

**Figure 1 ijerph-18-11564-f001:**
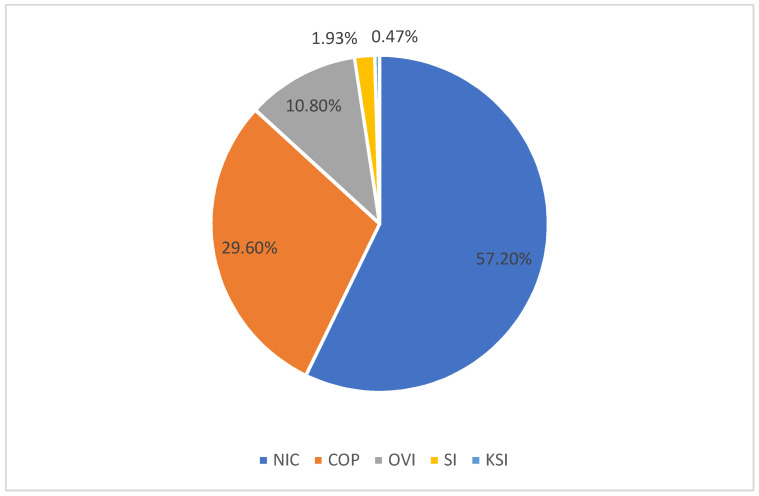
The proportion of crash injury severity in data. NIC—non-injury crash; COP—complaint of pain; OVI—other visible injury; SI—severe injury; KSI—killed.

**Figure 2 ijerph-18-11564-f002:**
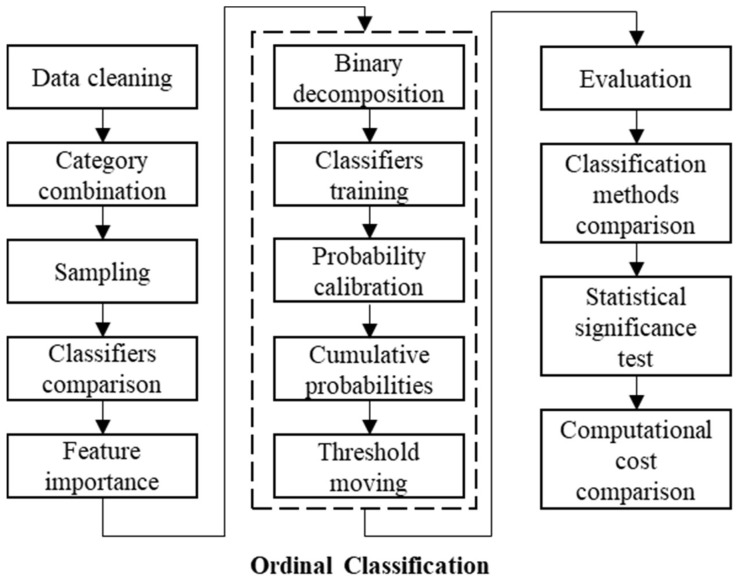
Research framework.

**Figure 3 ijerph-18-11564-f003:**
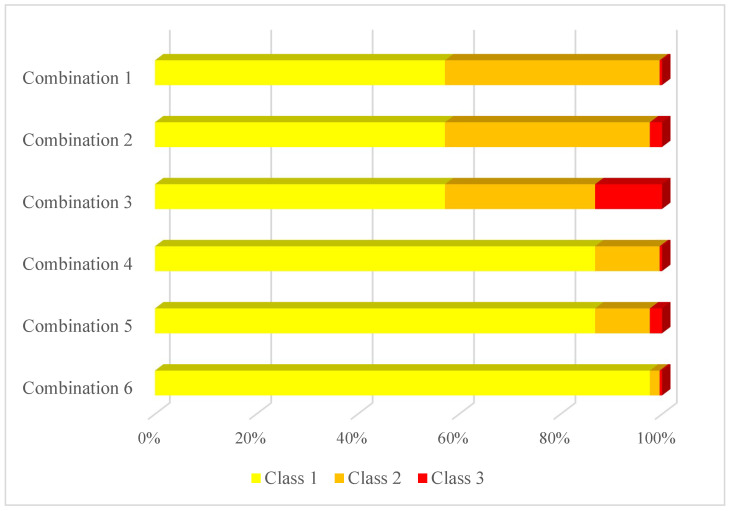
Six methods of 5-category combination.

**Figure 4 ijerph-18-11564-f004:**
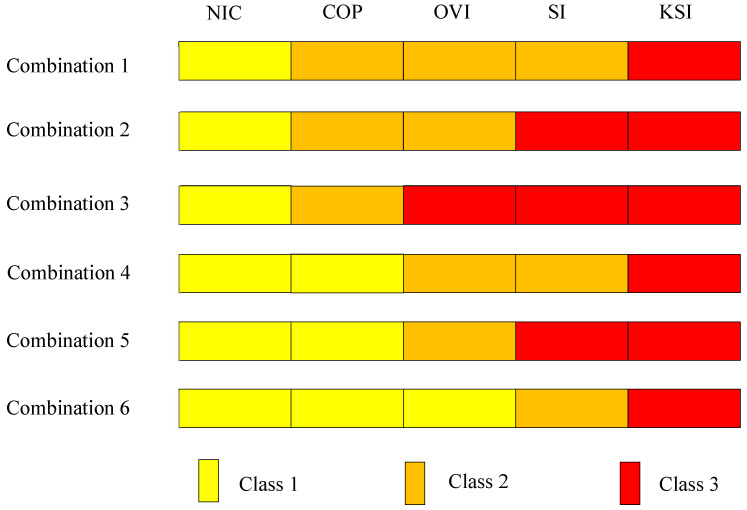
The proportion of classes in each method of categorizing.

**Figure 5 ijerph-18-11564-f005:**
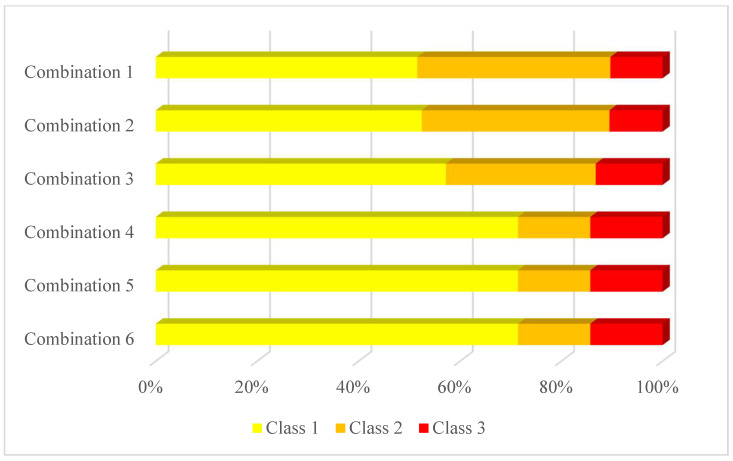
The proportion of classes in each method of category combination after sampling.

**Figure 6 ijerph-18-11564-f006:**
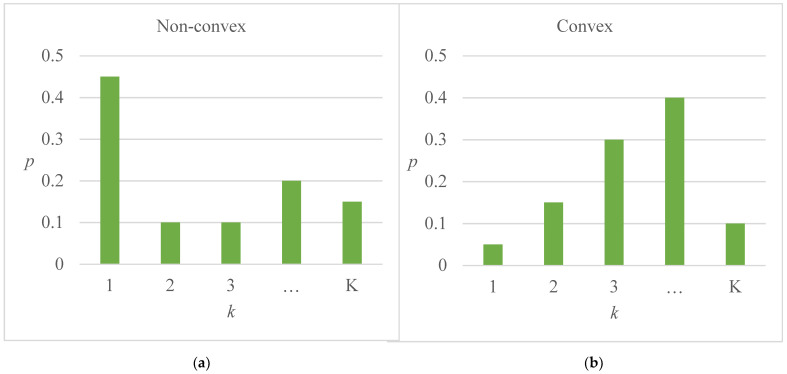
Illustration of rank inconsistency and rank consistency. (**a**) rank inconsistency; (**b**) rank consistency.

**Figure 7 ijerph-18-11564-f007:**
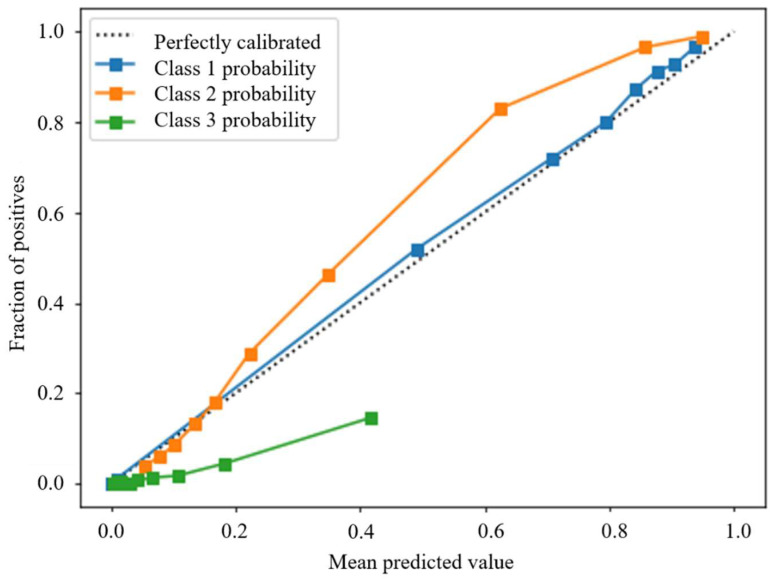
Probability plot of 3 classes before calibration.

**Figure 8 ijerph-18-11564-f008:**
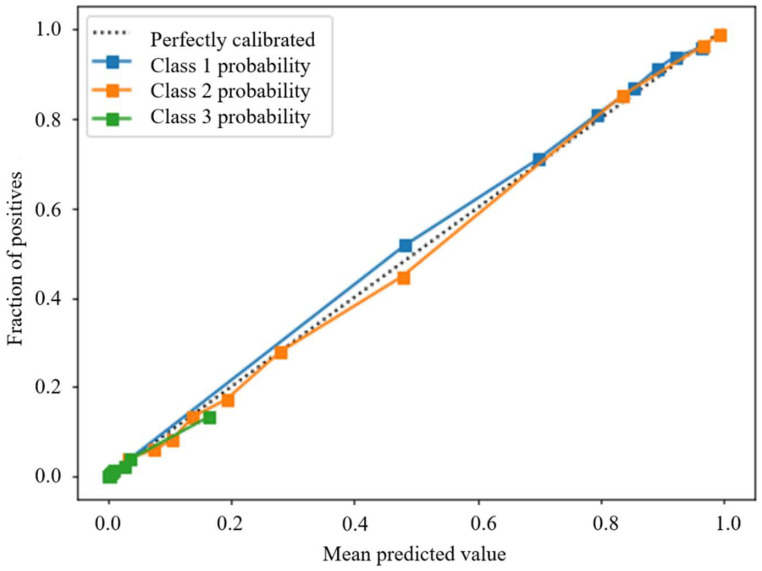
Probability plot of 3 classes after calibration.

**Table 1 ijerph-18-11564-t001:** Summary of ordinal classification methods.

Class Prediction Based on	Cumulative Binary Decomposition	One-vs-All Binary Decomposition
Class Probability	Frank’s method [22]	Beckham1 and Beckham2 [24]
Cumulative Probability	Cheng’s method [23]	This paper

**Table 2 ijerph-18-11564-t002:** Precisions of five-category classification problem.

Classifier	Precision (%)					Macro-Average
	NIC	COP	OVI	SI	KSI	
MLP	99.1	56.4	36.5	0.4	1.9	38.9
XGBoost	99.3	64.9	23.4	1.9	4.5	38.8
SVM	100	64.7	12.7	0	0	35.5

**Table 3 ijerph-18-11564-t003:** Precisions of three-class classification problem for six combinations before SMOTE-NC sampling.

Combination	Precision (%)			Macro-Average
	Class 1	Class 2	Class 3	
1	98.0	73.7	0.0	57.2
2	98.4	73.1	4.8	58.8
3	99.3	62.6	33.6	65.2
4	97.7	34.6	0.0	44.1
5	98.3	25.0	6.9	43.4
6	99.9	1.1	6.1	35.7

**Table 4 ijerph-18-11564-t004:** Performance of three-class classification problem for six combinations after SMOTE-NC sampling.

Combination	Group	Class 1	Class 2	Class 3	Macro-Average
1	Precision (%)	97.8	71.0	22.7	63.9
	Recall (%)	83.5	95.6	7.2	62.1
	F1 (%)	90.1	81.5	11.0	60.9
2	Precision (%)	98.0	67.8	27.5	64.4
	Recall (%)	83.3	92.6	21.4	65.8
	F1 (%)	90.1	78.3	24.1	64.2
3	Precision (%)	99.3	62.6	33.6	65.2
	Recall (%)	82.4	75.4	68.2	75.4
	F1 (%)	90.1	68.4	45.0	67.8
4	Precision (%)	97.1	23.1	36.4	52.2
	Recall (%)	91.2	66.4	5.6	54.4
	F1 (%)	94.1	34.3	9.7	46.0
5	Precision (%)	97.1	11.8	36.7	48.5
	Recall (%)	90.7	54.8	18.4	54.6
	F1 (%)	93.8	19.4	24.5	45.9
6	Precision (%)	97.0	21.6	27.3	48.6
	Recall (%)	98.4	18.6	8.7	41.9
	F1 (%)	97.7	20.0	13.2	43.6

**Table 5 ijerph-18-11564-t005:** Permutation feature importance for six combinations.

Variable	Definition	Combination					
		1	2	3	4	5	6
occupant	Occupant type	0.13	0.25	0.19	0.07	0.07	0.04
seat	Seating position	0.06	0.06	0.07	0.05	0.06	0.06
collision	Collision type	0.06	0.06	0.07	0.07	0.07	0.07
factor	First associated factor	0.05	0.04	0.05	0.06	0.06	0.06
cause	Primary collision cause	0.05	0.03	0.04	0.06	0.05	0.05
road	Roadway class	0.05	0.03	0.04	0.06	0.05	0.07
eject	Ejected from	0.10	0.09	0.11	0.02	0.11	0.09
object	First object struck	0.06	0.06	0.05	0.05	0.07	0.07
vehicles	number of vehicles	0.11	0.10	0.06	0.07	0.08	0.12
alcohol	alcohol involved	0.01	0.01	0.01	0.05	0.01	0.02
gender	driver’s gender	0.02	0.01	0.01	0.06	0.01	0.02
drv_safe	Driver safety equipment	0.05	0.06	0.06	0.10	0.07	0.06
occ_age	Occupant’s age	0.06	0.04	0.04	0.01	0.06	0.07
drv_age	Driver’s age	0.04	0.03	0.03	0.01	0.04	0.04
motor	Motorcycle involved	0.02	0.01	0.01	0.11	0.01	0.02
occ_safe	Occupant safety equipment	0.06	0.06	0.06	0.06	0.07	0.06
veh_year	Vehicle model year	0.08	0.07	0.11	0.11	0.11	0.08

**Table 6 ijerph-18-11564-t006:** Performance of six classification methods.

Method	Evaluation Metric	Class			Macro-Average
		1	2	3	
Nominal classification	Precision (%)	98.0	67.8	27.5	64.4
	Recall (%)	83.3	92.6	21.4	65.8
	F1 (%)	90.1	78.3	24.1	64.2
Beckham1	Precision (%)	92.9	73.6	22.7	63.0
	Recall (%)	85.6	83.9	21.8	63.8
	F1 (%)	89.1	78.4	22.3	63.2
Beckham2	Precision (%)	95.6	75.7	0.00	57.1
	Recall (%)	84.4	86.9	0.00	57.1
	F1 (%)	89.7	80.9	0.00	56.9
Frank	Precision (%)	97.6	68.2	31.6	65.8
	Recall (%)	83.6	92.1	23.6	66.4
	F1 (%)	90.1	78.4	27.0	65.1
Cheng	Precision (%)	96.0	70.3	29.2	65.2
	Recall (%)	84.4	88.9	24.1	65.8
	F1 (%)	89.8	78.5	26.4	64.9
This paper	Precision (%)	94.0	68.9	41.2	68.0
	Recall (%)	85.2	86.4	21.3	64.3
	F1 (%)	89.4	76.7	28.1	64.7

**Table 7 ijerph-18-11564-t007:** Paired Student’s *t*-test result.

Method A	Method B	*t*	*p*-Value
This paper	Nominal classification	24.99	0.000
	Beckham1	15.82	0.000
	Beckham2	12.68	0.000
	Frank	4.89	0.005
	Cheng	16.58	0.000

**Table 8 ijerph-18-11564-t008:** Performance of proposed method with different thresholds.

Method	Evaluation Metric	Class			Macro-Average
		1	2	3	
*T*_1_ = 0.5 *T*_2_ = 0.5	Precision (%)	86.4	81.5	15.8	61.2
	Recall (%)	88.2	77.4	27.3	64.3
	F1 (%)	87.3	79.4	20.0	62.2
*T*_1_ = 0.43 *T*_2_ = 0.33	Precision (%)	94.0	68.9	41.2	68.0
	Recall (%)	85.2	86.4	21.3	64.3
	F1 (%)	89.4	76.7	28.1	64.7

## Data Availability

Data can be requested to download at https://www.hsisinfo.org/ (accessed on 1 November 2021).

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
