# Peer review of "Crash Injury Severity Prediction Using an Ordinal Classification Machine Learning Approach"

_ijerph, 2021, doi:10.3390/ijerph182111564_

Round 1

Reviewer 1 Report

The article is very interesting and contains very interesting research. Analyzing the causes and predicting the occurrence of road accidents is important all over the world. Nevertheless, it is necessary to make some corrections.

In the abstract, it is necessary to precisely formulate the aim of the study. Many researchers, when reading an abstract, decide whether to the reads article.

Also in the introduction you need to carefully write the aim of the study. It results from the content, but I propose to formulate this additionally.

I think that all formulas should be numbered and not placed directly in the text. Please check the formulas for errors.

The graphic material is sufficient and legible.

Chapter 4. Results and Discussion cannot contain the word "Discussion" because discusion is a reference to the research of other authors, which is not here. I think the "results" alone are enough.

The presented analysis of the literature is sufficient.

Reviewer 2 Report

Thank you for submitting this manuscript for review. It’s interesting to propose a novel ordinal classification approach to predict traffic crash injury severity due to the negative impacts of traffic crash. However, in its current format, the manuscript is not suitable for publication. I don't think the the approach is novel enough, although to your knowledge, this is the first paper to predict traffic crash injury severity using real-world crash data with ordinal classification machine learning. Separately, the ordinal classification machine learning algorithms are existing by far, and the method of dealing with unbalanced data is also existing. The novelty of this paper is a bit confusing and needs to be reorganized. Maybe the application of this method or the novelty of the attempt can be mentioned instead of methodology itself. In addition, I note comments pertaining to relevant sections below.

In section 1, the too specific limitations of existing studies have been mentioned many times,for example, most existing machine learning methods applied in crash severity prediction treat crash severity levels as nominal data without order information. But it still doesn't directly explain the research significance of this article. Too specific limitations are not sufficiently convincing, nor can they prove novelty. I would expect the authors to complete the research significance when considering order information in prediction of crash injury severity.

In section 2, the data set itself has 5 categories, why integrated into fewer categories, please give a reasonable explanation.

Feature engineering in section3 is needed to expand, what are the selection criteria for the 17 variables? And in model selection, why choose neural network, XGBoost, and SVM classifiers? Please provide some additional information.

Finally, there are some details in this paper, as shown in Figure 1, whether the specific ratio can be marked? And the process of determining model parameters can be briefly provided. Generally speaking, the ideas in this article are very good, and the workload is large. I look forward to your changes.

Reviewer 3 Report

I think the authors have done a good job in describing the crash injury severity by introducing a novel ordinal classification machine learning approach. I only have a few comments that the authors may pay attention to further improve the quality of the paper:

1) resolution of the figure is low, please consider improving the resolution of those figures;

2) lines 539-542, please provide the support for the optimal values of T1 and T2. Also, the precision rate of class 3 is 15.8%, I did not find this number, please double check this number.

Round 2

Reviewer 2 Report

I am satisfied with the revisions. No further comments.